# Slip bursts during coalescence of slow slip events in Cascadia

Quentin Bletery [1✉] & Jean-Mathieu Nocquet [1,2]

Both laboratory experiments and dynamic simulations suggest that earthquakes can be preceded by a precursory phase of slow slip. Observing processes leading to an acceleration or spreading of slow slip along faults is therefore key to understand the dynamics potentially leading to seismic ruptures. Here, we use continuous GPS measurements of the ground displacement to image the daily slip along the fault beneath Vancouver Island during a slow slip event in 2013. We image the coalescence of three originally distinct slow slip fronts merging together. We show that during coalescence phases lasting for 2 to 5 days, the rate of energy (moment) release significantly increases. This observation supports the view proposed by theoretical and experimental studies that the coalescence of slow slip fronts is a possible mechanism for initiating earthquakes.

[1] Université Côte d'Azur, IRD, CNRS, Observatoire de la Côte d'Azur, Géoazur, 250 rue Albert Einstein, 06560 Valbonne, France. [2] Institut de Physique du Globe de Paris, Université de Paris, CNRS, 75238 Paris, France. ✉email: bletery@geoazur.unice.fr

In western North America, the northern Cascadia subduction fault experiences slow slip events (SSEs) approximately every 14 months[1]. As slow slip develops, it generates a particular type of micro-earthquakes, known as low-frequency earthquakes that superpose as tectonic tremors[1–3]. The space–time correlation between slow slip and tremors[4,5] gives the opportunity to study the details of the evolution of SSEs, taking advantage of independent measurements of the ground displacement and tremor activity[6–12]. Here we focus on a complex SSE that occurred beneath Vancouver Island in September–October 2013. We use Global Positioning System (GPS) time series of the ground displacement to image the daily evolution of slip along the Cascadia subduction fault (Figs. 1 and 2) in relation to the daily distribution of tremors cataloged by the Pacific Northwest Seismic Network (PNSN)[13,14].

## Results

**Two coalescence episodes.** We find that slow slip initiates in the area of Seattle on September 7, 2013, associated with intense tremor activity (Supplementary Fig. 3). During the first 2 weeks, most of the slip and tremors remain confined to the same 60 × 30 km$^2$ area, with a maximum activity on September 22 (Supplementary Figs. 3 and 4). Starting on September 15, tremors also emerge ~250 km from that area at latitude 50° under Vancouver Island with low-amplitude slip sparsely detected from September 19 onward (Supplementary Fig. 4). Tremor locations indicate a southeastward, along-strike, and constant depth migration at ~10 km per day. On September 26, slip catches up with tremors and extends southeastward. On September 24, an additional area of tremors arises between the two SSE areas, in the southern part of Vancouver island, soon followed by slow slip. On September

**Fig. 1 First coalescence episode.** Color maps show the slow slip rate inverted along the fault on each day from September 25 to September 28, 2013. Blue dots show the location of tremors recorded on the corresponding day. Isolines indicate the fault depth from 20 to 80 km.

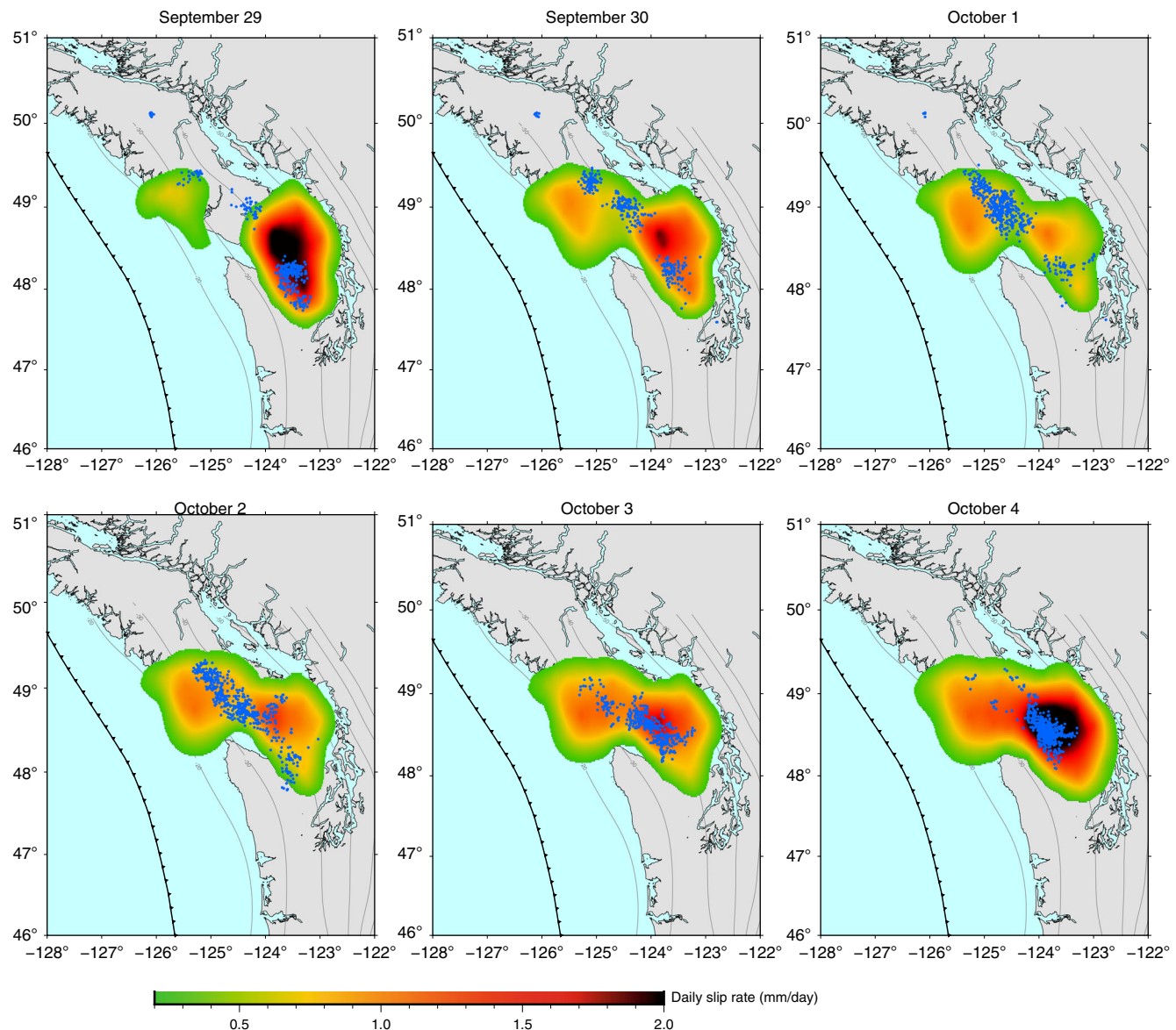

**Fig. 2 Second coalescence episode.** Color maps show the slow slip rate inverted along the fault on each day from September 29 to October 4, 2013. Blue dots show the location of tremors recorded on the corresponding day. Isolines indicate the fault depth from 20 to 80 km.

26, we see three clearly distinct patches of slip, each of them associated with a tremor swarm (Fig. 1). From September 27, the region located in between the southern and central patches starts slipping. On September 28, the two southern patches are indistinguishable one from the other (Fig. 1). During this merging phase, the moment rate release raises by a factor 10 in 3 days and the daily maximum slip increases by a factor 3 in 2 days (Fig. 3a). As an independent check for this acceleration, we find that the average daily incremental displacement recorded by the GPS stations close to the merging area is multiplied by a factor 8 (Fig. 3b) (see "Methods—Edge filter"). An interesting aspect is that, before the merging phase, the slow slip at the southern patch was fading out (see decrease in moment rate release after September 22, Fig. 3a), suggesting that the coalescence of the two previously distinct slipping areas is the actual cause for the observed increase in moment rate release. Following the first merging, the moment release slightly diminishes (Figs. 2 and 3a). A second merging phase occurs quickly afterwards (Fig. 2). This latter merging phase is also associated with a significant pulse in the moment release rate (2× between October 1 and October 4),

which eventually decreases to end the sequence on October 12 (Fig. 3, Supplementary Fig. 5).

**Interaction between distant slip fronts?** The 2013 Fall sequence highlights two successive coalescence episodes of pre-existing distinct slipping areas. Though jumping and halting are common in SSEs[6], merging episodes similar to this sequence have not been observed during other SSEs in Cascadia or elsewhere. Nevertheless, the coalescence of nearby slipping areas might not be uncommon. In a rate-and-state framework[15,16], as slip propagates from one area toward another, it progressively meets increasing positive shear stress generated by the counterpart patch, which in turn enhances its slip and promotes further propagation, until both slipping areas coalesce. For both merging phases during the 2013 sequence, our inversion suggests that coalescence initiates when the distance between the two SSE areas is on the order of one length-scale of their size. However, interaction at larger distance cannot be ruled out since tremors —and by extension SSEs—are known to be sensitive to extremely small stress

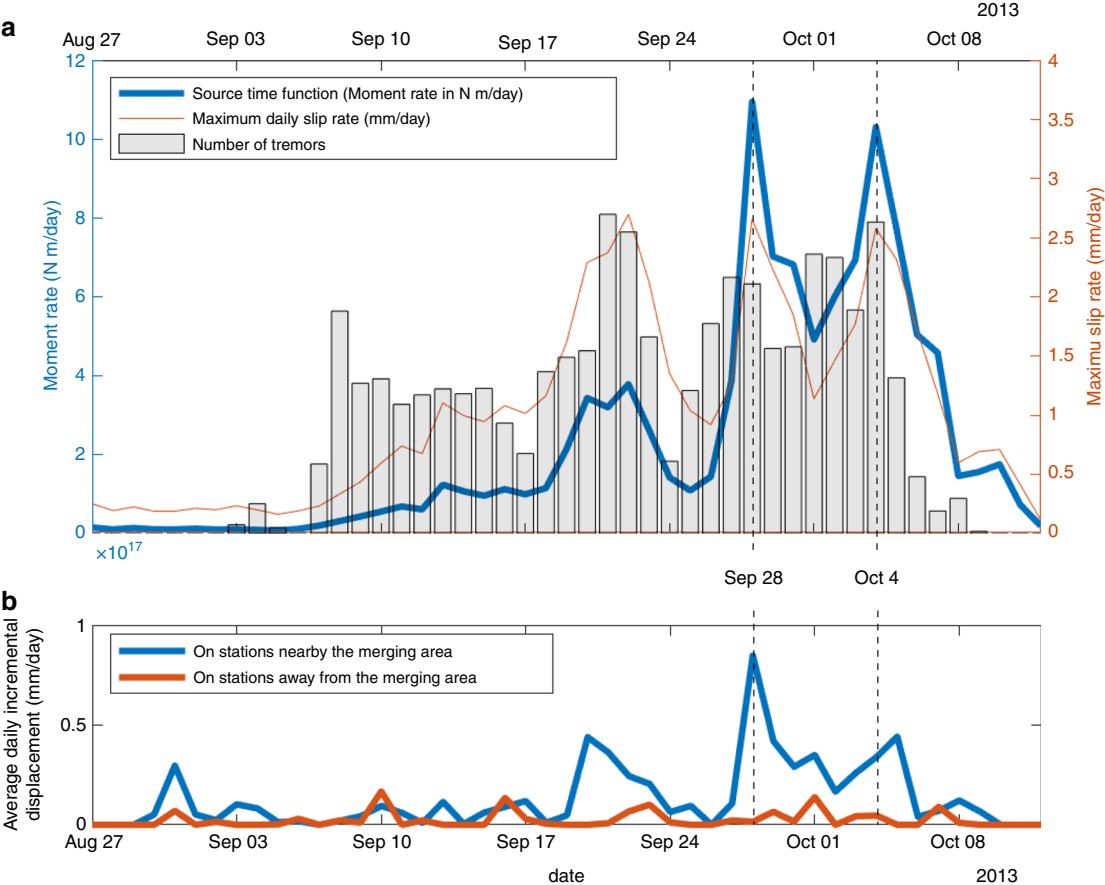

**Fig. 3 Increase in moment release during both coalescence episodes. a** Evolution of the moment rate release during the entire sequence. Gray histograms indicate the relative number of tremors on each day. The red thin curve shows the maximum daily slip on the fault every day. The blue thick curve shows the moment rate release. **b** Average daily displacement recorded on GPS stations close to (blue curve) and away from (red curve) the merging area after application of an edge filter, highlighting an increase in GPS data amplitude within the stations located close to the coalescence episodes. Stations used in both stacks are shown in Supplementary Fig. 1 (blue and red stations close to and away from the merging area, respectively).

perturbations, such as those generated by tides[17]. The southern migration of the northern slip front or the onset of the center one possibly reflects such long-distance interaction.

**Enhanced slip during coalescence episodes.** Both merging phases are associated with a burst in moment rate release resulting from a combination of a growth of the slipping area filling the space between the two patches and an increase in daily slip velocity (Fig. 3a). Moment release bursts result in an acceleration of the daily incremental displacement at GPS sites located above the merging area, directly seen in the GPS data (Fig. 3b). The period of merging between September 27 and October 5 (9 days) contributes to more than half (59%) of the total moment released during the entire sequence (47 days). Furthermore, peak moment rates on September 28 and October 4 represent a third (33%) of the moment released during the merging period. These ratios suggest a merging-induced burst-like behavior of slip at a daily time scale. Slip might, however, experience sub-daily variations. Indeed, analyses of seismic data indicate that micro-seismic activity, presumably driven by slow slip, is clustered in minute-to-hour-long bursts[18,19]. The daily incremental slip inferred here is likely an average of multiple shorter episodic events, with the implication that slip rates may punctually be significantly faster than the daily slip velocities inferred from our inversion.

## Discussion

Laboratory-derived constitutive friction laws predict that, for certain combinations of parameters referred to as velocity-weakening regimes, when sliding initiates, the friction drops proportionally to the logarithm of the sliding velocity, which in turn leads to enhanced sliding velocity[15,16]. Applied to the earthquake initiation problem, dynamic simulations have shown that the conditions required for slip to become unstable are functions—among other parameters—of the slip rate or stress perturbation in the nucleation area[15,16]. The coalescence of slow slip fronts therefore provides a mechanism countering the usual damping of slip observed during SSEs. This mechanism promotes instability on the fault through a slip acceleration and a stress increase at the tip of the locked area, which is larger than the summed contribution of individual fronts.

Based on laboratory experiments and numerical simulation, Dieterich intuited 40 years ago that slip velocity must increase if two previously independent zones of slip approach and coalesce to form a single zone[20]. His reasoning was that, because stress on a dislocation is controlled by the ratio of slip over the length of the slipping area, the rate of slip must accelerate to double the displacement as the segments merge to form a single zone. Dieterich[20] and more recent studies[21,22] proposed the coalescence of slow slip fronts as a possible mechanism to initiate earthquakes. While not the case here, the mechanism might have been at play during SSEs that have been proposed to trigger earthquakes.

Among the examples of recent large earthquakes that have been preceded by SSEs[23–25], multiple slow slips accompanied by intense seismicity have been documented from months to days before the $M_w$ 8.1 April 1, 2014 Iquique earthquake[26,27]. Although the merging of SSEs has not been geodetically identified, the sequence shows some similarities with the 2013 Cascadia sequence. The migration of small earthquakes and activity of repeating earthquakes both suggest the existence of two slow slip areas separated by ~120 km in January and February 2014 north and south of the epicenter. While the northern area appears to be stationary and shows repeating earthquake activity until March 23, clustered and intense seismicity following the March 16 $M_w$ 6.7 earthquake migrates from south to north[27], leaving the possibility for the two slow slip areas to have merged near the future epicenter.

The 2013 Cascadia sequence provides an observational evidence of a phenomenon previously proposed by physical models[20–22]. It demonstrates that periods of SSE merging are key periods during which slip is enhanced, possibly bringing the fault closer to seismic failure. In the quest for earthquake precursors, it provides a clue on where to pay attention.

## Methods

**GPS data**. We use 59 GPS daily time series from the final Geodesy Advancing Geosciences and EarthScope combined solution[28] available from the UNAVCO data center and added 3 stations (ELIZ, WOST, GLDR) from the Pacific Northwest Geodetic Array of the Central Washington University, north of Vancouver island where coverage is sparse. The location of the 62 used stations is shown in Supplementary Fig. 1. We remove outliers and seasonal signals over the whole time series. We detrend the time series, so that the velocity during the weeks before and after the 2013 event is zero. This procedure ensures that the transient displacement recorded at the GPS sites reflects positive reverse slip at the plate interface. Regional common mode motion evaluated by stacking time series for 10 sites with complete data and good repeatability located 100–300 km away for the slip area was found to be negligible. The obtained time series are shown in Supplementary Figs. 7–17 and discussed in Supplementary Note 1.

**Green's functions**. We use a curved geometry for the subduction interface based on the slab2.0 model for Cascadia[29] discretized into 307 quasi-equilateral triangular subfaults, ranging from 46°N to 50°N in latitude and from the trench down to 50 km. The rake is calculated on each subfault accounting for the rotation of the North American plate with respect to the Juan de Fuca plate (Euler rotation pole : −111.7°E, 32.0°N[30]). We calculate the transfer matrix g relating unit slip in the rake direction at each triangular subfault to the displacement components at the GPS sites using the solution for triangular dislocation elements in a uniform elastic half space[31].

**Slip time-dependent inversion**. We invert for daily incremental slip from August 27, 2013 to October 12, 2013 using only the horizontal components. Our approach relates the vector of cumulative displacements $d_k$ at GPS sites since August 27 at date k to the sum of all daily incremental slip $\delta s_i$ before date k multiplied by the Green's function g:

$$\sum_{i=1}^{k} g \delta s_i = d_k \qquad (1)$$

Repeated for all dates k, Eq. (1) leads to a linear system $\mathbf{G}\delta\mathbf{s} = \mathbf{d}$, where $\mathbf{G}$ is a block-triangular matrix made of g, $\mathbf{d}$ and $\delta\mathbf{s}$ are the displacement vector and modeled daily slip resulting from concatenating all $d_k$ and $\delta s_i$, respectively. The chosen formulation allows to solve consistently for the daily slip at every subfault in a single inversion. Two important aspects of our inversion procedure are a non-negativity constraint[32] that proscribes backward slip increment at all time steps, strongly reducing the solution space[33] and the absence of temporal smoothing that allows a better restitution of slip acceleration. Spatial regularization constraints are imposed by the mean of a model covariance matrix controlling the level of damping and smoothing with respect to an a priori model[8,34,35] here taken as 0. The model covariance matrix is taken as an isotropic decreasing exponential[8]:

$$C_{m i,j} = \left( \sigma \frac{d_0}{D_c} \right)^2 \exp(-d_{i,j}/D_c) \qquad (2)$$

where $C_{m i,j}$ are the model covariance matrix elements corresponding to the ith and jth subfaults, $\sigma$ a constant (unit in mm day$^{-1/2}$) controlling the weight of the regularization, $d_{i,j}$ the distance between the center of subfaults i and j, and $D_c$ is a correlation length controlling the level of spatial smoothing[36]. $d_0$ is a reference distance taken as the mean length of the triangles. No temporal smoothing is added here aside from the non-negativity constraint imposing that slip must grow through time. We used $D_c = 50$ km, a value on the order of the distance between the center of two adjacent subfaults. We tested different values of $\sigma$ and plotted the gain in misfit reduction as a function of regularization ($\sigma$). We obtained a figure commonly known as an L-curve (Supplementary Fig. 2). We chose $\sigma = 3$ mm day$^{-1/2}$, the point at the corner of the L-curve for which the misfit minimization becomes marginal when regularization increases.

**Tremor data**. We analyze the obtained slip distribution in light of the tremor activity (Figs. 1–3, Supplementary Figs. 3–6). We use the tremor catalog from the PNSN[13,14]. Because GPS data are daily averages of the positions measured from 00:00:00 to 23:59:30 UTC, they can be seen as the positions at 12:00:00 UTC every day. Therefore, the daily slip on the fault we image on day i should be seen as the slip from day $(i − 1)$ at 12:00:00 to day i at 11:59:59 (Figs. 1 and 2, Supplementary Figs. 3–5). Accordingly, the tremors represented on the subfigure of day i are tremors recorded from day $(i − 1)$ at 12:00:00 to day i at 11:59:59 (Figs. 1–3, Supplementary Figs. 3–5).

**Moment rate release**. The moment rate release (Fig. 3a) is calculated by summing the contributions of all subfaults assuming a uniform elastic Earth with an elastic shear modulus of 30 GPa. The cumulative moment is $1.10 \times 10^{19}$ N m, equivalent to a moment magnitude $M_w = 6.6$, the maximum cumulative slip is 35.3 mm, the maximum slip rate 2.7 mm day$^{-1}$. Because the dimensions of all the subfaults are equal, the moment rate release is strictly proportional to the average slip rate integrated over the ruptured surface. The daily number of tremors correlates fairly well with the moment release rate (coefficient of correlation: 0.63) and even better with daily maximum slip rate (coefficient of correlation: 0.76) (Fig. 3a). Furthermore, the two merging phases are associated with spikes in tremor activity (Figs. 1–3) and the day-to-day spatial correlation between slow slip and the independently inferred distribution of tremors (Figs. 1 and 2, Supplementary Figs. 3–5, Supplementary Movie 1) suggests that the main features of our inverted slip distribution are robust. Additional resolution tests can be found in Supplementary Information in the online version of the article (Supplementary Note 2, Supplementary Figs. 18–22).

**Edge filter**. In order to verify that the moment rate increase found in our inversion does not arise from fitting noise, we investigate the high-frequency properties of our time series. We apply a one-dimensional edge filter[37] to the north and east components of the GPS displacement time series. Edge filtering is a technique widely used in image processing that allows to retrieve sharp gradients in noisy data. We differentiate the filtered time series, take the norm, stack the obtained time series, and normalize it by the number of sites. The obtained time series reflect the average displacement over 1 day. We apply this procedure independently to a selection of 16 sites located in the area of the mergings (stations represented in blue in Supplementary Fig. 1) and to 10 sites away from the merging area in order to assess the level of noise (stations represented in red in Supplementary Fig. 1). We verify that sites located away from the SSE show spatially averaged daily velocity <0.2 mm day$^{-1}$ (Fig. 3b, red curve). On the contrary, sites located near the merging area show a clear increase in recorded slip displacement during the periods of enhanced tremor activity. In particular, the late-September merging phase is associated with a sharp (8×) increase in daily displacement (Fig. 3b, blue curve). This analysis demonstrates, independently from the inversion, that the 2013 sequence had several periods of coherent displacement acceleration, the largest one corresponding to the period of SSE merging around September 28 as found in our inversion.

## Data availability

The GPS time series and tremor catalog we used are respectively available on the UNAVCO (https://www.unavco.org) and PNSN websites (https://tremor.pnsn.org).

## Code availability

The pyacs library we used for data processing and for the kinematic slip inversion is available upon request to J.-M.N. (email: nocquet@geoazur.unice.fr).

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

## Acknowledgements

We built our fault geometry based on the slab2.0 model available online at the following address: https://www.sciencebase.gov/catalog/item/5aa1b00ee4b0b1c392e86467/. We thank A. Wech, K. Creager, and PNSN for letting their tremor catalog available (https://tremor.pnsn.org). We downloaded the GPS time series from the UNAVCO website (https://www.unavco.org). We used the Generic Mapping Tools GMT5 (https://gmt.soest.hawaii.edu/). We thank Jean-Paul Ampuero and Martin Vallée for helpful discussion. This work has been supported by the French government, through the UCAJEDI Investments in the Future project managed by the National Research Agency (ANR) ANR-15-IDEX-01 and by the ANR research grants E-POST, REMAKE, and S5.

## Author contributions

Q.B. had the original idea. J.-M.N. provided numerical tools (PYACS python library) and expertise. Both authors contributed to the analysis, interpretation and to the manuscript preparation.

## Competing interests

The authors declare no competing interests.
