## [Peer Review File · Nature Communications]

Reviewers' comments:

Reviewer #1 (Remarks to the Author):

This is a well-written and illustrated paper that addresses one of the hottest topics today in Geophysics, namely the extent to which slow slip events act as pre-cursors to large and great subduction zone earthquakes, and the relevant underlying physics. The results described by the authors represent an important advance in our understanding of these complex events. In my opinion, it should be published.

I have one minor point that the authors may wish to address. The basic thesis of the paper is that merging of SSEs leads to acceleration, which can lead to a large earthquake. A simplistic reading of the paper leads to the opposite conclusion - they have observed merging and acceleration, but a large earthquake did not occur. I am sympathetic to the authors' viewpoint that this is merely an example of a class of event that COULD lead to an earthquake, but some more nuanced discussion might be useful. For example, can they speculate on what the key threshold for an earthquake is based on their results? Perhaps it is peak slip rate, combined with region of the seismogenic zone being late in the earthquake cycle. If peak slip rate is the key, then merging of events may not be required - just acceleration (by any means) such some key threshold is crossed). From their figures it appears that peak slip rate was about 2 mm/day (the exact number should be referenced in the paper). Can they compare this value to other studies (where earthquakes did or did not occur). I believe there have been observations of this in both Central America and Japan.

It may also be useful to plot maximum daily slip rate on Figure 3, it is presumably closely related to moment rate.

END

Reviewer #2 (Remarks to the Author):

The paper highlights that slow slip fronts progressively merged together during a 2013 sequence along Cascadia subduction zone, and the slip rate temporally increased during the merging phases. Coalescence of rupture front has been proposed numerically and in laboratory experiments, but there has never been a field observation. If the coalescence of slow slip fronts is well constrained by geodetic data, the present result will be really exciting to the scientific community of seismology and geodesy. The manuscript is well organized and concise. However, there are some concerns/weakness related to the slip evolution inverted from the geodetic measurements as described below. Especially, the reliability and robustness of the inverted slip are not convincing to me. These points should be clarified and revised properly before the publication.

1) To confirm the reliability of each slip patch, it is required to add maps showing displacement vectors at each time step (daily). Especially, the increment displacement vectors at each time step are essentially important before and after the merging phase of two different slip patches. The space and time evolution of horizontal displacement vectors are basic information to judge the reliability of slip distribution.

2) I cannot understand well the method to invert daily slip amount on the plate interface. I guess that the authors inverted the cumulative displacement vectors on one day and the successive day, respectively, to obtain a spatial distribution of the cumulative slip on the plate interface. Then, the authors subtract the two cumulative slip amounts on the plate interface. Is it correct? More detailed explanation point should be added. And, estimation error of slip usually increases by subtraction. What is the minimum resolvable slip amount? Is a central slip-patch with small displacement on 26 September (Fig. 1) really constrained?

- 3) The fitted curves to each displacement time series are very smooth, as shown from Figure S7 to S16. Why are these fitted curves so smooth? It seems to be quite strange.
- 4) From Figure S7 to S16, many displacement signals are not well fitted by the predicted curves (e.g., CLRS, ALBH, PTAL, PTRF, SC02, SC04, P440, ...). Why doesn't it work so much? Especially, the predicted amplitude is often smaller than that from the observation.
- 5) From Figure S7 to S16, there are offsets from the starting day as seen in many stations (e.g., P691, P441..). Why? The detrending does not work well, isn't it?
- 6) How did you account for common mode errors as well as seasonal variations? The explanation about this point should be added.
- 7) Around merging area of two slow slip patches, there are several bore-hole strain meters. Can you find out the similar acceleration of slip rate from the bore-hole strain measurements at the same timing? The bore-hole data can strengthen your results.
- 8) In Figure 3, the moment rate release has the second peak with a significant delay (3-4 days) after the merging of southern patch with northern one. Do you have an interpretation?
- 9) In several references (Kaneko and Ampuero, 2011 (<http://dx.doi.org/10.1029/2011GL049953>.); Fukuyama et al., 2018 (<https://doi.org/10.1016/j.tecto.2017.12.023>))., coalescence of rupture front has been discussed numerically and in laboratory experiments. These papers have suggested the rupture front coalescence may initiate the unstable ruptures.

Reviewer #3 (Remarks to the Author):

Review Summary:

The authors propose that if a slow slip event (SSE) nucleates at multiple locations or fronts on a subduction plate interface, then at the time of merging of these slip fronts into a single slip zone, the slip accelerates, thereby defining areas and periods of enhanced probability for the occurrence of large earthquakes. This hypothesis is not novel, having been presented in a theoretical context by researchers such as J.H. Dieterich (Ref. #2). What is novel and interesting both to the seismic community and the wider field is this paper's detailed study of a SSE that occurred along the northern Cascadia Subduction Zone which demonstrates, for the first time, that such slip merging can indeed occur.

However, the authors' contention that this merging of slip results in an observed increase in slip rate is open to question. Referring to Figure 3, the authors argue that the dramatic increase in moment rate release implies an increase in average slip rate, which I interpret to refer to a temporal change in the slip velocity. Is it possible that changes in moment rate release reflect changes in the area of the fault that is active at a given time? Authors need to discuss the contributions of changes in slip velocity versus the contributions of changes in slip area to the daily moment rate release. If the total area of active fault slip on a given day is the main contributor to variations in moment rate release, then authors should also keep in mind that low-amplitude, distributed slip would not be well resolved, especially in mid- to northern Vancouver Island where the spatial coverage of GPS stations is sparse.

In addition, the authors themselves point out that the slip velocities must be much greater (closer to seismic velocities) in order to trigger seismic rupture. If such high slip velocities are required to

increase the likelihood of an earthquake, does it not render the (relatively) modest proposed change in slip velocity caused by merging of slips far less significant in determining earthquake probability? Authors need to address this in order to justify the speculation that it is the coalescence of slip fronts that is the significant factor.

More detailed comments:

Abstract L1/2

Replace "earthquake ruptures are preceded" with "earthquake ruptures can be preceded"

Abstract L10

"slip drastically accelerates" See "Review Summary"

Page 1 L1/2

insert "northern" before "Cascadia subduction fault". The southern portion of Cascadia has different return periods for SSE (Brudzinski & Allen, *Geology* (2007) 35 (10): 907-910)

Page 2 L16/17

"Both slip and tremor locations indicate a southeastward ...migration"

This migration is not apparent for the slip. On Figure S4, Sep.16 to Sep.24, the small patch of northern slip looks essentially stationary. This is likely controlled by the fact that there is only a single GPS station (NTKA) in this area.

Page 2 L24/25

"...meaning that the average slip rate increased by a factor > 5"

What is meant here by "average slip rate"? The total area identified to be involved in slip appears to have increased by roughly a factor of 5 from Sep.26 to Sep.28. Can this be the the main cause of the sharp increase in moment rate release shown in Figure 3? How are changes in moment rate release due to slip area variations seperated from changes due to slip velocity variations?

Page 3 L9

"Although such [coalescence] episodes have not been observed during other SSE's in Cascadia"

Have the authors themselves examined other SSE's in Cascadia or are they stating that no-one else has pointed them out? (which should not be surprising since the concept of actual observed SSE slip coalescence is the novel aspect of this paper). The authors are referred to a JGR paper by Honn et al., 2009, Northern Cascadia episodic tremor and slip: A decade of tremor observations from 1997 to 2007, *J. Geophys. Res.*, 114, B00A12, doi:10.1029/2008JB006046. This paper proposes a classification scheme for northern Cascadia ETS (Episodic Tremor and Slip) episodes and points out that "Halting and jumping are very common in ETS migration patterns, and along-strike migration can happen in both directions." An examination of the slip behaviour over a decade of observations presented by Kao et al., in the context of slip coalescence, may provide additional supporting data.

Page 4 L18/19

"The scale and duration of the possible micro-events composing SSE's..." By pointing out that there could exist local and episodic slip rates much larger than the resolved daily average slip rates, are the authors implying that slip rates closer to seismic slip rates are required for an SSE to trigger an earthquake? If so, then the changes in slip rate resulting from the convergence of SSE's is not a key factor in determining the probability of the occurrence of a large earthquake. Instead, it would be the generation of high-amplitude, short-duration micro-slip events which would govern earthquake likelihood. If this is the case, then the authors need to show that the coalescence of slow slip fronts results in a greater number of possibly critical micro-slips.

Supplementary Materials

Figure S1

GPS coverage in central to northern Vancouver Island is sparse, limiting the resolution of the slip inversion. There are 4 GPS stations in this region that, unfortunately, are not included in the nominal UNAVCO analyses (see edited version of Figure S1). This raises the question: Does the mismatch in spatial sampling density between the northern and southern areas present any intrinsic bias in defining the extent and development of the slip zones?

Figure S18

In evaluating the resolution power of their approach it should be pointed out that the question is not the fidelity of reproducing "target" slips but one of uniqueness. How many other model slip distributions will adequately replicate the target slips within nominal observational error?

Dear reviewers,

Please find below our point-to-point response to your comments.

Reviewer #1 (Remarks to the Author):

This is a well-written and illustrated paper that addresses one of the hottest topics today in Geophysics, namely the extent to which slow slip events act as pre-cursors to large and great subduction zone earthquakes, and the relevant underlying physics. The results described by the authors represent an important advance in our understanding of these complex events. In my opinion, it should be published.

I have one minor point that the authors may wish to address. The basic thesis of the paper is that merging of SSEs leads to acceleration, which can lead to a large earthquake. A simplistic reading of the paper leads to the opposite conclusion - they have observed merging and acceleration, but a large earthquake did not occur. I am sympathetic to the authors' viewpoint that this is merely an example of a class of event that COULD lead to an earthquake, but some more nuanced discussion might be useful.

We adapted the manuscript by stressing on the actual observation (i.e. the increase in moment rate release in response to SSE merging) and nuancing the more speculative interpretation on the potential link with large earthquake initiation.

For example, can they speculate on what the key threshold for an earthquake is based on their results? Perhaps it is peak slip rate, combined with region of the seismogenic zone being late in the earthquake cycle. If peak slip rate is the key, then merging of events may not be required - just acceleration (by any means) such some key threshold is crossed).

We preferred to avoid speculations. The robust observation that we provide is that merging phases of previously distinct SSEs lead to a drastic increase in moment rate release. In that sense, the coalescence of SSE counteracts the processes contributing to damp the slip during SSEs, but we do not believe that we could go further than this qualitative argument.

From their figures it appears that peak slip rate was about 2 mm/day (the exact number should be referenced in the paper).

The peak slip rate was indeed 2 mm/day. It is now 2.7 mm/day in the inversion updated with the additional sites suggested by reviewer #3. We added the number in the manuscript and now display the time series of maximum slip rate during the studied period in Fig 3.

Can they compare this value to other studies (where earthquakes did or did not occur). I believe there have been observations of this in both Central America and Japan.

Kato et al. (Science, 2011) linked the nucleation of the M_w 9.0 Tohoku earthquake (Japan, 2011) to changes in slip rate on the order of a few cm/day, one order of magnitude faster than the slip rate we infer. Voss et al. (Science Adv., 2017) suggested that a slow slip event with similar slip rate than what we find triggered an M_w 7.6 earthquake in Nicaragua (2012). Radiguet et al. (Nat. Geo., 2013) documented an SSE in Mexico preceding an M_w 7.3 earthquake (Papanoa, 2014) but did not estimate a slip rate. While these comparisons are interesting, we preferred not to mention them in the manuscript to avoid defining loosely-constrained quantitative thresholds, which likely depend on many properties specific to each location and time in the seismic cycle.

It may also be useful to plot maximum daily slip rate on Figure 3, it is presumably closely related to moment rate.

Done

END

Reviewer #2 (Remarks to the Author):

The paper highlights that slow slip fronts progressively merged together during a 2013 sequence along Cascadia subduction zone, and the slip rate temporally increased during the merging phases. Coalescence of rupture front has been proposed numerically and in laboratory experiments, but there has never been a field observation. If the

coalescence of slow slip fronts is well constrained by geodetic data, the present result will be really exciting to the scientific community of seismology and geodesy. The manuscript is well organized and concise. However, there are some concerns/weakness related to the slip evolution inverted from the geodetic measurements as described below. Especially, the reliability and robustness of the inverted slip are not convincing to me. These points should be clarified and revised properly before the publication.

We now provide a more thorough description of our methodology in the Method section.

1) To confirm the reliability of each slip patch, it is required to add maps showing displacement vectors at each time step (daily). Especially, the increment displacement vectors at each time step are essentially important before and after the merging phase of two different slip patches. The space and time evolution of horizontal displacement vectors are basic information to judge the reliability of slip distribution.

Intrinsic noise in GPS time series of the order of 1-2 mm prevents to extract the differential displacement between two consecutive days. In order to answer your question, we investigated the relationship between the sudden moment rate increase found in our inversion and the property of the GPS time series. Identifying significant displacement between two successive days on a single station from raw data being hopeless, we filter the data both in time and space. For time filtering, we used an edge filter that preserves the sharpest gradient in noisy data. We apply the filter on the East and North components for every GPS station. Space filtering is simply obtained by stacking the filtered time series for sites expected to show the largest signal, here close to the merging area (see Fig. S1). We then differentiate the filtered time series and take the norm of it. The same procedure is also applied using sites located away from the merging area to quantify the level of remaining noise. Both stacks are normalized by the number of sites used to allow a direct comparison. This approach highlights that moment rate increases noted around September 22, 28 and October 4 have their origin in a drastic increase of the displacement rate at GPS sites close to the merging area, with the September 28 period being the most important in norm and the shortest in time (Fig. 3b). See also response to next comment.

2) I cannot understand well the method to invert daily slip amount on the plate interface. I guess that the authors inverted the cumulative displacement vectors on one day and the successive day, respectively, to obtain a spatial distribution of the cumulative slip on the plate interface. Then, the authors subtract the two cumulative slip amounts on the plate interface. Is it correct? More detailed explanation point should be added. And, estimation error of slip usually increases by subtraction. What is the minimum resolvable slip amount? Is a central slip-patch with small displacement on 26 September (Fig. 1) really constrained?

We acknowledge that we did not provide enough explanation about our methodology. Because your question 1 and 2 are closely related to methodology and resolution we answer them jointly.

The methodology we use is a full time-dependent finite-fault slip inversion that allows to solve for the slip on every day slip on every subfault in a single and consistent inversion. The way we build the linear system relating the daily slip to the displacement time series can be understood as follow:

After 1 day, the vector of displacement at the GPS site d_1 is the product of slip s_1 during day 1 with the elastic Green's tensor g :

$$g s_1 = d_1$$

After 2 days, the vector of displacement at the GPS sites d_2 is the sum of the slip s_1 during day 1 and s_2 during day 2 multiplied by g :

$$g (s_1 + s_2) = d_2$$

After n days, we have $g (s_1 + s_2 + \dots + s_n) = d_n$

The previous equation can be organized in a single linear system:

$$\begin{bmatrix} g & 0 & \cdots & 0 & \cdots & 0 \\ g & g & \cdots & 0 & \cdots & 0 \\ \vdots & \vdots & \ddots & \vdots & \cdots & \vdots \\ g & g & \cdots & g & \cdots & 0 \\ \vdots & \vdots & \vdots & \vdots & \ddots & \vdots \\ g & g & \cdots & g & \cdots & g \end{bmatrix} \begin{bmatrix} s_1 \\ s_2 \\ \vdots \\ s_k \\ \vdots \\ s_n \end{bmatrix} = \begin{bmatrix} d_1 \\ d_2 \\ \vdots \\ d_k \\ \vdots \\ d_n \end{bmatrix}$$

which is then solved using non-negative least-squares and adding spatial regularization constraints. The described approach therefore solves simultaneously for all daily slip at all dates directly from the GPS time series. In other words, our formulation balances between respecting the cumulative displacement observed at GPS sites through time and their change from one day to another.

Daily displacements at GPS sites include both noise intrinsic to GPS and signal coherent both in space and time coming from slip on the fault. The level of noise in GPS time series (1-2 mm) prevents to reliably extract daily displacement from one day to the next, but our inversion scheme is able to extract the signal that is coherent among several GPS sites and which is consistent with growing slip (ensured by the non-negativity constraint) at the fault.

With this approach, the slip inverted on a given subfault for a given day must be consistent with the slip inverted on any other patch on any other day so that (1) the sum of daily slip multiplied by the Green's tensor fits (as well as possible) the full GPS time series, and (2) the daily slip is positive, an assumption consistent with our a priori knowledge of the underlying physics (i.e. no backward slip is allowed). The non-negativity constraint in the kinematic inversion considerably reduces the solution space and acts as a natural filter which allows us to extract tiny features of the slip distribution and its evolution.

Although marginal probability density functions can theoretically be computed for this inversion scheme (see Nocquet, 2018 for semi-analytical solutions), the high number of unknown parameters here (14429) prevents such calculation in practice. We are therefore unable to answer quantitatively your question of whether the central small slip patch is resolved or not. Small isolated patches add roughness to the model and increase the cost function such that the model tends to prefer single patch rather than having separated ones. Furthermore, the distribution of tremors independently indicates that some slow slip must occurs in that area on the days we image the central patch. Finally, the synthetic test presented in the supplements gives a sense of the resolution expected for our inversion both for time, space and moment rate retrieval.

The method section has been modified to describe more thoroughly our inversion approach, with the constraint of the limited space allowed by Nature Communication.

3) The fitted curves to each displacement time series are very smooth, as shown from Figure S7 to S16. Why are these fitted curves so smooth? It seems to be quite strange.

Our inversion scheme only models the part of the time series that are coherent in space and time. The non-negativity constraint which prevents backward slip and therefore strongly limits the possibility of non-monotonicity of the fitted curves, acts as a natural filter. Obtaining modeled time series that are smoother than the original data is rather a good sign that we are not modeling noise as SSE signal.

In order to answer your question more precisely, we apply the procedure we applied the recorded time series (Fig 3.b) to the time series predicted by our model (Fig R1 below, yellow curve for the merging area, purple curve for the sites away from the merging area). The figure below shows that the modeled time series are actually not as smooth as they visually appear, and also include some increase in the displacement rate. The filtering effect of the inversion can be seen for sites away from the merging area where displacement rate (and no displacement) is found. This result suggests that our inversion probably underestimate the moment rate increase around September 28.

Figure R1: Same as Fig. 3b in the main text (blue and red curves) enriched with equivalent plots for the predicted time series close to (yellow) and away from (purple) the merging area.

4) From Figure S7 to S16, many displacement signals are not well fitted by the predicted curves (e.g., CLRS, ALBH, PTAL, PTRF, SC02, SC04, P440, ...). Why doesn't it work so much? Especially, the predicted amplitude is often smaller than that from the observation.

Most of this pattern has been fixed in the revised version of the inversion, after improving the reference level for our time series (see response to next comment). The obtained fit in our updated inversion is overall fairly good.

5) From Figure S7 to S16, there are offsets from the starting day as seen in many stations (e.g., P691, P441...). Why? The detrending does not work well, isn't it?

Our inversion scheme requires a start date used as a reference to evaluate the displacement since that reference date. In the previous inversion, we did not pay attention on this point and for some sites, the inversion misfit was reflecting the misfit of the point used for reference with respect to the better fit of the whole time series. We corrected this issue by defining the reference position as the median of the first 8 days (August 27th – September 3rd) during which no SSE signal is noticeable. No obvious offset is now visible after applying this change (see Figs S7-S17). Furthermore, this change has no impact on the inversion results.

6) How did you account for common mode errors as well as seasonal variations? The explanation about this point should be added.

In our processing, we do not remove a regional common mode in the usual sense of evaluating a stack of regional residual time series (for sites not impacted by the signal) and then removing it to all time series. For every time series taken individually, we estimated a trend and seasonal signal so that 4 weeks before and after the SSE, the slope is null. This procedure ensures that only the signal related to the SSE is modeled.

To answer your question, we evaluate whether this procedure could leave a possible common mode motion that might bias our results. We computed the stack of time series for the sites away from the location of the SSE and for which no signal is predicted from the inversion. The obtained stacked time series are shown below in Fig R2. Figure R2 shows that there is no obvious trend during the period used for the inversion and no obvious signal during the merging period. The root mean square of the common mode time series is small (0.4 mm for both the East and North components). Finally, our test using the edge filtering does not show common pattern between sites located close and far away from the SSE (Fig R1, 3.b). We therefore conclude that our processing of the GPS time series is adequate to represent the displacement induced by the SSE.

Figure R2: regional common mode estimated from the sites distant from the SSE (red stations in Fig. S1).

7) Around merging area of two slow slip patches, there are several bore-hole strain meters. Can you find out the similar acceleration of slip rate from the bore-hole strain measurements at the same timing? The bore-hole data can strengthen your results.

We are aware of the difficulty of reliably extracting tectonic signal from borehole strainmeter data, which include tidal loading, effects, and atmospheric pressure changes components (e.g. Hawthorne & Rubin, JGR, 2010, 2013). We carefully inspect all components for all borehole data available. Sharp peaks of strain on the order of 1 microstrain are seen on September 28-30 and a smoother anomaly of several days around October 4 are visible in the uncorrected data of several sites (B001, B943, B013, B943). However, most of these anomalies disappear in the data corrected (PBO level 2 products) from tidal and pressure effects.

In an effort to extract a possible signal associated with the merging, we show below the strain time series for several sites close to the merging area with no data gap (Figure R3). The time series have been detrended and normalized according to their standard deviation. Anomalous behavior is observed during the period of maximum tremor and slip activity for the southern patch around September 22, with a larger anomaly around September 28, reversing in early October. We made a preliminary attempt to separate different contributions to the signal using an Independent Component Analysis (ICA). Results for a decomposition using 4 components is shown in Figure R4. Component 1 & 2 appear to isolate anomalies related to period of enhanced tremor activity and slip. Although these results are encouraging, we are unable to demonstrate that these anomalies are related to the SSE acceleration. We do not believe they provide additional robust evidence of period of enhanced moment rate during the 2013 SSE and prefer not mentioning them in the manuscript.

Figure R3: Time series of borehole strain data (uncorrected). Eee and Enn are the strain components along East and North respectively. Time series have been detrended and normalized by their standard deviation.

Figure R4: Independent Component Decomposition of the strain time series shown in Fig R3 for borehole B007, B011 & B013. Note that the first two components show anomalies synchronous to period of enhanced slip and tremor activity. The reversal of signal in component 2 from late September to early October, also seen in the data might reflect a migration of slip with respect to the borehole location.

8) In Figure 3, the moment rate release has the second peak with a significant delay (3-4 days) after the merging of southern patch with northern one. Do you have an interpretation?

It looks from Fig 2 that while the first contact is made on September 30 for the 2nd merging, 2 distinct (red) patches are still distinguishable until October 4. The tremor distribution also suggests that the merging is progressive and is only fully achieved on October 4, coinciding with the spike in moment rate release.

9) In several references (Kaneko and Ampuero, 2011 (<http://dx.doi.org/10.1029/2011GL049953>); Fukuyama et al., 2018 (<https://doi.org/10.1016/j.tecto.2017.12.023>)), coalescence of rupture front has been discussed numerically and in laboratory experiments. These papers have suggested the rupture front coalescence may initiate the unstable ruptures.

References added in the abstract and the discussion.

Reviewer #3 (Remarks to the Author):

Review Summary:

The authors propose that if a slow slip event (SSE) nucleates at multiple locations or fronts on a subduction plate interface, then at the time of merging of these slip fronts into a single slip zone, the slip accelerates, thereby defining areas and periods of enhanced probability for the occurrence of large earthquakes. This hypothesis is not novel, having been presented in a theoretical context by researchers such as J.H. Dieterich (Ref. #2). What is novel and interesting both to the seismic community and the wider field is this paper's detailed study of a SSE that occurred along the northern Cascadia Subduction Zone which demonstrates, for the first time, that such slip merging can indeed occur.

However, the authors' contention that this merging of slip results in an observed increase in slip rate is open to question. Referring to Figure 3, the authors argue that the dramatic increase in moment rate release implies an increase in average slip rate, which I interpret to refer to a temporal change in the slip velocity. Is it possible that changes in moment rate release reflect changes in the area of the fault that is active at a given time? Authors need to discuss the contributions of changes in slip velocity versus the contributions of changes in slip area to the daily moment rate release. If the total area of active fault slip on a given day is the main contributor to variations in moment rate release, then authors should also keep in mind that low-amplitude, distributed slip would not be well resolved, especially in mid- to northern Vancouver Island where the spatial coverage of GPS stations is sparse.

We added the evolution of the daily maximum slip on Fig 3.a (red curve). Comparison between the blue and red curves in Fig 3.a gives a sense of how much of the increase in moment rate release is due to slip acceleration and how much is due to spreading of the slipping area. Although the maximum slip rate is sensitive to the regularization constraints (moment rate is far less sensitive), our inversion finds that the daily maximum slip is multiplied by 3 during September 26 and September 28. Given the fact that the surface encompassing the two patches had a limited growth at the time of the merging, this suggests that at least part of the increase in moment rate release is due to slip acceleration.

In addition, the authors themselves point out that the slip velocities must be much greater (closer to seismic velocities) in order to trigger seismic rupture. If such high slip velocities are required to increase the likelihood of an earthquake, does it not render the (relatively) modest proposed change in slip velocity caused by merging of slips far less significant in determining earthquake probability? Authors need to address this in order to justify the speculation that it is the coalescence of slip fronts that is the significant factor.

We agree with your comment that the slip velocities would have to be greater in order to trigger seismic rupture. As answered to reviewer #1 comment, the robust observation that we provide is that merging phases of previously distinct SSEs lead to a drastic increase in moment rate release. In that sense, the coalescence of SSE counteracts the processes contributing to damp the slip during SSEs. Although there are certainly many other factors controlling the bifurcation towards unstable slip, we still believe that enhanced moment rate periods during SSE would define more favorable condition for slip to run away.

More detailed comments:

Abstract L1/2

Replace "earthquake ruptures are preceded" with "earthquake ruptures can be preceded"

Done

Abstract L10

"slip drastically accelerates" See "Review Summary"

Changed to "the rate of energy (moment) release significantly increases"

Page 1 L1/2

insert "northern" before "Cascadia subduction fault". The southern portion of Cascadia has different return periods for SSE (Brudzinski & Allen, *Geology* (2007) 35 (10): 907-910)

Done

Page 2 L16/17

"Both slip and tremor locations indicate a southeastward ...migration". This migration is not apparent for the slip. On

Figure S4, Sep.16 to Sep.24, the small patch of northern slip looks essentially stationary. This is likely controlled by the fact that there is only a single GPS station (NTKA) in this area.

True. We updated the manuscript accordingly.

Page 2 L24/25

"...meaning that the average slip rate increased by a factor > 5 "

What is meant here by "average slip rate"? The total area identified to be involved in slip appears to have increased by roughly a factor of 5 from Sep.26 to Sep.28. Can this be the the main cause of the sharp increase in moment rate release shown in Figure 3? How are changes in moment rate release due to slip area variations seperated from changes due to slip velocity variations?

What we meant by average slip rate here was moment rate release (which is equal to the slip integrated over the entire fault on a given day times the shear modulus, which is taken constant so that moment release rate is strictly proportional to the average slip rate).

The evolution of the daily maximum slip rate (red curve in Fig 3.a) indicates that the fastest slip velocity is multiplied by 3 from September 26 to September 28. This provides the rough estimate that half of the increase in moment rate release is due to slip acceleration and half of it is due to spreading of the slipping area.

Page 3 L9

"Although such [coalescence] episodes have not been observed during other SSE's in Cascadia" Have the authors themselves examined other SSE's in Cascadia or are they stating that no-one else has pointed them out? (which should not be surprising since the concept of actual observed SSE slip coalescence is the novel aspect of this paper). The authors are referred to a JGR paper by Honn et al., 2009, Northern Cascadia episodic tremor and slip: A decade of tremor observations from 1997 to 2007, J. Geophys. Res., 114, B00A12, doi:10.1029/2008JB006046. This paper proposes a classification scheme for northern Cascadia ETS (Episodic Tremor and Slip) episodes and points out that "Halting and jumping are very common in ETS migration patterns, and along-strike migration can happen in both directions." An examination of the slip behaviour over a decade of observations presented by Kao et al., in the context of slip coalescence, may provide additional supporting data.

To the best of our knowledge, no coalescence of slow slip events was pointed out before our study. Reference added page 3.

Page 4 L18/19

"The scale and duration of the possible micro-events composing SSE's..." By pointing out that there could exist local and episodic slip rates much larger than the resolved daily average slip rates, are the authors implying that slip rates closer to seismic slip rates are required for an SSE to trigger an earthquake? If so, then the changes in slip rate resulting from the convergence of SSE's is not a key factor in determining the probability of the occurrence of a large earthquake.

It is one factor, which could be key in some cases (e.g. faster original slip, later stage in the seismic cycle).

Instead, it would be the generation of high-amplitude, short-duration micro-slip events which would govern earthquake likelihood. If this is the case, then the authors need to show that the coalescence of slow slip fronts results in a greater number of possibly critical micro-slips.

This cannot be done with present geodetic instrumentation. We agree that this point was rather speculative and revised the manuscript accordingly.

Supplementary Materials

Figure S1

GPS coverage in central to northern Vancouver Island is sparse, limiting the resolution of the slip inversion. There are 4 GPS stations in this region that, unfortunately, are not included in the nominal UNAVCO analyses (see edited version of Figure S1). This raises the question: Does the mismatch in spatial sampling density between the northern and southern areas present any intrinsic bias in defining the extent and development of the slip zones?

We included WOST, ELIZ and GLDR (we could not find trace of MYRA) in our analysis. These stations do not show any obvious signal related to the SSE sequence but improve the resolution in the northern part. The result of the inversion was not significantly changed.

It is correct to say that the resolution is much better on the southern part than on the northern part of the rupture. It is also much better inland than offshore. The resolution test reveals that the northern section is "more under-predicted" than southern part (Fig S20), which is likely a consequence of sparser station coverage, which supports your comment. Resolution is never homogeneous in finite-fault inversion problems. In the absence of observational constraints, the inversion will tend to smooth the slip distribution, meaning that the sparse distribution of GPS stations in the northern likely results in over-smoothing of the recovered slip in this area relative to the southern better-instrumented part.

Figure S18

In evaluating the resolution power of their approach it should be pointed out that the question is not the fidelity of reproducing "target" slips but one of uniqueness. How many other model slip distributions will adequately replicate the target slips within nominal observational error?

This is extremely difficult to evaluate, and in our case computationally intractable. Bayesian approaches allow to make such estimations (e.g. Bletery et al., JGR, 2016, Nocquet, GJI, 2018) but they currently cannot be used for inversion of more than a few hundred parameters because computing time increases exponentially with the number of inverted parameters. Since we need here to perform a kinematic inversion to recover the slip every day on each patch, the number of parameters we invert for is 307 subfaults times 47 days = 14,429, which currently cannot be handled by Bayesian inversion approaches.

REVIEWERS' COMMENTS:

Reviewer #1 (Remarks to the Author):

The authors have done a good job of responding to the reviewer comments. In my opinion, the manuscript is ready for publication.

Reviewer #2 (Remarks to the Author):

According to the comments, the authors properly revised the manuscript including the supplementary file. Although the manuscript is almost close to the publication, several minor revisions as follow are still needed.

Abstract:

"the view proposed by theoretical studies (4-6)"

->

"the view proposed by theoretical and experimental studies (4-6)"

The title of Supplementary Materials is different from that in the main manuscript. The authors changed the title in the revised manuscript but forgot to change it in the supplement.

Figure S1: The authors selected 10 sites away from the merging area in order to assess the level of noise in the edge filter analysis (red triangles). How did you select these red sites? There are jumping of selections, not selected in an area as like blue sites. At least, it is required to explain the reason.

Figure S21: The recovered slip rate is shown by a thick dashed line. But the broken line spacing is too coarse to see the precise temporal change in the slip rate. You need the modification.

Dear Editor,

Please find below our response to the last comments of the reviewers on our manuscript.

Reviewer #1 (Remarks to the Author):

The authors have done a good job of responding to the reviewer comments. In my opinion, the manuscript is ready for publication.

Reviewer #2 (Remarks to the Author):

According to the comments, the authors properly revised the manuscript including the supplementary file. Although the manuscript is almost close to the publication, several minor revisions as follow are still needed.

Abstract: “the view proposed by theoretical studies (4–6)” -> “the view proposed by theoretical and experimental studies (4–6)”

Done

The title of Supplementary Materials is different from that in the main manuscript. The authors changed the title in the revised manuscript but forgot to change it in the supplement.

Done

Figure S1: The authors selected 10 sites away from the merging area in order to assess the level of noise in the edge filter analysis (red triangles). How did you select these red sites? There are jumping of selections, not selected in an area as like blue sites. At least, it is required to explain the reason.

A first selection of sites has been made so that they are located outside the slow slip area (see Figure S6) and with no signal indicated by our modeling results (Figure S14). Within that selection, we kept GPS sites with no missing data within the studied period (see Figure S14, sites P408, P417, P425) and removed some additional sites with obvious outliers (LINH, CHWK) or large daily scatter (e.g. P691) unrelated to SSE, that would have triggered false detection in the edge filtering analysis.

Figure S21: The recovered slip rate is shown by a thick dashed line. But the broken line spacing is too coarse to see the precise temporal change in the slip rate. You need the modification.

Done

Reviewer #3 (Remarks to the Author):

I have looked at the revised paper by Bletery and Nocquet and their responses to concerns raised by previous reviewers. In my opinion the authors have done a good job of responding to the comments, strengthening what was already a good paper.

Reviewer #3 raised several issues related to data limitations and the smoothing that is inherent in the inversion process that allows estimation of processes at depth based on surface data affected by noise. While it is always useful to keep these limitations on mind, I believe the authors have done a reasonable job of analyzing their data and interpreting it, without over-interpretation. While the data set they are

using is not perfect (obviously no data set is) it is certainly more robust than most, and as the authors point out, allows some very interesting observations that are highly relevant to current research on stress triggering of earthquakes. I believe their results should be published.